# Seven Additional Patients with *SOX17* Related Pulmonary Arterial Hypertension and Review of the Literature

**DOI:** 10.3390/genes14101965

**Published:** 2023-10-20

**Authors:** Natalia Gallego-Zazo, Lucía Miranda-Alcaraz, Alejandro Cruz-Utrilla, María Jesús del Cerro Marín, María Álvarez-Fuente, María del Mar Rodríguez Vázquez del Rey, Inmaculada Guillén Rodríguez, Victor Manuel Becerra-Munoz, Amparo Moya-Bonora, Nuria Ochoa Parra, Alejandro Parra, Patricia Pascual, Mario Cazalla, Cristina Silván, Pedro Arias, Diana Valverde, Vinicio de Jesús-Pérez, Pablo Lapunzina, Pilar Escribano-Subías, Jair Tenorio-Castano

**Affiliations:** 1Instituto de Genética Médica y Molecular (INGEMM), Instituto de Investigación del Hospital Universitario La Paz (IdiPaz), Hospital Universitario La Paz, 28046 Madrid, Spain; luciamiranda.ingemm@gmail.com (L.M.-A.); alejandro.parra.externo@salud.madrid.org (A.P.); patripascual0@gmail.com (P.P.); mario.cazalla16@gmail.com (M.C.); cristina_sf8@hotmail.com (C.S.); palajara@gmail.com (P.A.); plapunzina@gmail.com (P.L.); 2CIBERER, Centro de Investigación Biomédica de Enfermedades Raras en Red, Instituto de Salud Carlos III, 28029 Madrid, Spain; 3ERN-ITHACA, European Reference Network on Rare Malformations Syndromes, Intellectual and Other Neuro-Developmental Disorders, 75019 Paris, France; 4Unidad Multidisciplinar de Hipertensión Pulmonar, Servicio de Cardiología, Hospital Universitario 12 de Octubre, 28041 Madrid, Spain; acruzutrilla@gmail.com (A.C.-U.); nuriaochoaparra@hotmail.com (N.O.P.); pilar.escribano.subias@gmail.com (P.E.-S.); 5ERN-LUNG, European Reference Network on Rare Lung Diseases (Pulmonary Hypertension), 60596 Frankfurt am Main, Germany; 6CIBERCV, Centro de Investigación Biomédica en Red de Enfermedades Cardiovasculares, Instituto de Salud Carlos III, 28029 Madrid, Spain; vmbecerram@gmail.com; 7Unidad de Hipertensión Pulmonar Pediátrica, Servicio de Cardiología Pediátrica, Hospital Universitario Ramón y Cajal, Instituto de Investigación Biomédica del Hospital Universitario Ramón y Cajal (Irycis), 28034 Madrid, Spain; majecerro@yahoo.es (M.J.d.C.M.); m.alvarez.fuente@salud.madrid.org (M.Á.-F.); 8Unidad de Cardiología Pediátrica, Hospital Universitario Virgen de las Nieves, 18014 Granada, Spain; marrvr@gmail.com; 9Unidad de Cardiología Pediátrica, Hospital Universitario Virgen del Rocío, 41013 Sevilla, Spain; miguillenr@hotmail.com; 10Unidad de Gestión Clínica Área del Corazón, Instituto de Investigación Biomédica de Málaga (IBIMA), Hospital Universitario Virgen de la Victoria, Universidad de Málaga, 29590 Málaga, Spain; 11Unidad de Cardiología Pediátrica, Departamento de Pediatría, Hospital Universitario La Fe, 46026 Valencia, Spain; amparmoya@gmail.com; 12Centro de Investigación en Nonomateriais e Biomedicina (CINBIO), Universidad de Vigo, 36310 Vigo, Spain; dianaval@uvigo.es; 13Instituto de Investigación Sanitaria Galicia Sur, Hospital Álvaro Cunqueiro, 36310 Vigo, Spain; 14Centro de Investigaciones Biomédicas (CINBIO), 36310 Vigo, Spain; 15Division of Pulmonary and Critical Care Medicine, Department of Medicine, Stanford University, Stanford, CA 94305, USA; vdejesus@stanford.edu

**Keywords:** pulmonary arterial hypertension, *SOX17*, whole exome sequencing, genomic medicine, personalized medicine, cardiovascular disorders, congenital heart disease

## Abstract

Pulmonary arterial hypertension (PAH) is an infrequent disorder characterized by high blood pressure in the pulmonary arteries. It may lead to premature death or the requirement for lung and/or heart transplantation. Genetics plays an important and increasing role in the diagnosis of PAH. Here, we report seven additional patients with variants in *SOX17* and a review of sixty previously described patients in the literature. Patients described in this study suffered with additional conditions including large septal defects, as described by other groups. Collectively, sixty-seven PAH patients have been reported so far with variants in *SOX17*, including missense and loss-of-function (LoF) variants. The majority of the loss-of-function variants found in *SOX17* were detected in the last exon of the gene. Meanwhile, most missense variants were located within exon one, suggesting a probable tolerated change at the amino terminal part of the protein. In addition, we reported two idiopathic PAH patients presenting with the same variant previously detected in five patients by other studies, suggesting a possible hot spot. Research conducted on PAH associated with congenital heart disease (CHD) indicated that variants in *SOX17* might be particularly prevalent in this subgroup, as two out of our seven additional patients presented with CHD. Further research is still necessary to clarify the precise association between the biological pathway of *SOX17* and the development of PAH.

## 1. Introduction

Group one of Pulmonary Hypertension (PH), or pulmonary arterial hypertension (PAH) [MIM 265400], is an infrequent and progressive precapillary disorder that affects the small arteries and capillaries in the lungs, leading to increased resistance to blood flow which can cause right heart pressure overload. At the molecular level, the mechanisms and biological pathways that drive the development of PAH are not yet fully understood. However, it is known that genetic predisposition, combined with environmental risk factors, plays an important role in its development. Exposure over time to certain epigenetic and environmental triggers can lead to progressive pulmonary vascular dysfunction, and these processes can be accelerated in individuals with variants of PAH-associated genes [1].

PAH is also classified into different groups, with idiopathic PAH (IPAH) being the most common. Other forms of PAH can be associated with other conditions such as connective tissue diseases (PAH-CTD), congenital heart defects (PAH-CHD), or drug and toxins exposure (PAH-TX) [2]. Although *BMPR2* (bone morphogenetic protein receptor 2) was the first gene described and is the most frequently mutated in heritable PAH (HPAH), representing around 60% of cases [3], nowadays, more than 20 genes have been associated with the development of PAH [4]. While there is no cure for PAH, early diagnosis of PAH is crucial; therefore, genetic screening is mandatory and can support clinical management and patients’ follow-up. For example, there is an increasing development of a new generation of drugs, such as Sotatercept^®^, which has recently demonstrated overt clinical benefits across multiple efficacy end points in patients with idiopathic or associated PAH [5]. The molecular mechanisms acting through Sotatercep^®^ include the regulation of the TGF-β pathway via the modulation of BMPR2 receptor activation. This highlights the importance of genomic advances toward to new drug development.

In recent years, genetics has become a powerful and very useful tool to confirm the clinical diagnosis of patients with PAH and has demonstrated the usefulness of its study in the design of new drugs, such as Sotatercept^®^, for the treatment of this incurable disease [6].

Herein, we will focus on the relationship between *SOX17* variants and the development of PAH. The causal relation between heterozygous *SOX17* variants and PAH was first proposed in 2018 by genome-wide gene burden tests, which enabled the identification of novel gene variants via a comparison of variant frequencies between patients with PAH and controls in the UK NIHR PAH cohort. Authors detected statistically significant enrichment of rare deleterious variants in this gene among patients, compared to controls [6]. Since then, 46 variants in 63 patients have been reported so far.

According to the ClinGen classification, *SOX17* is definitively associated with autosomal dominant PAH [7] (Table 1).

*SOX17* mutation carriers have a significantly younger age at diagnosis, putatively suggesting early onset PAH; however, this observation requires validation in larger patient cohorts. A whole exome sequencing study involving 256 patients with PAH strongly implicated *SOX17* pathogenic variants as a major risk factor in PAH-associated congenital heart disease and also provided independent validation for their role in IPAH [8]. Therefore, *SOX17* was included in our custom genetic screening for patients with suggestive PAH diagnosis. We identified seven patients with variants in *SOX17* not reported previously, which confirmed the initial PAH diagnosis, and we also reviewed both the clinical and molecular characteristics of our cohort and all the patients reported in the literature so far.

## 2. Materials and Methods

### 2.1. Patients

This study was approved by the ethical committee of Hospital Universitario La Paz (CEIC-PI1210) and Hospital Universitario 12 de Octubre (CEIm-21/484). In addition, informed consent was obtained from all patients and/or legal tutors. Patients were selected from the Spanish PAH Registries: REHAP (Spanish Adult Pulmonary Arterial Hypertension Registry) (https://www.rehap.org/, accessed on 4 September 2023) and REHIPED (Pediatric Pulmonary Hypertension Patient Registry) (https://www.rehiped.org/, accessed on 4 September 2023). For inclusion in the registry, the diagnosis of PAH required a right heart catheterization (RHC), with mean pulmonary artery pressure (mPAP) ≥25 mmHg, pulmonary vascular resistance (PVR) ≥3 WU, and pulmonary artery wedge pressure (PAWP) ≤15 mmHg. In this study, we included patients with the diagnosis of idiopathic, heritable, and toxin-induced PAH, as well as sporadic and heritable PVOD. The study period was from 2011 up to September 2023. In total, 1230 adult and pediatric patients with different etiologies were analyzed (Figure 1).

From this cohort population, we included all patients who presented with a *SOX17* pathogenic variant after genetic screening and collected retrospectively their clinical, hemodynamical and molecular features, and their outcomes. All patients were diagnosed following the clinical parameters established in diagnostic 2022 ERS/ESC guidelines [2]. Three out of the ninety-eight pediatric patients analyzed were identified as having a *SOX17* variant, representing 3.06% (3/98) of the pediatric cohort. From the adult cohort, we analyzed 1132 patients, and we found variants in 0.03% of the cohort (4/1132).

In addition to the seven patients reported from our cohort, we have summarized and reviewed the clinical and molecular features of all patients reported in the literature.

### 2.2. Genetic Analysis

In our cohort, we performed whole exome sequencing (WES). Library preparation was carried out using the Agilent SureSelect M (v6.0) All Exon Kit followed by sequencing in a NovaSeq6000 Sequencer (Illumina, San Diego, CA, USA), following manufacturer’s instructions. Variant prioritization was performed according to a custom algorithm by applying the VarSeq software v2.3.0 (Golden Helix, Bozeman, MT, USA) in order to detect single nucleotide variants (SNVs), InDels, and copy number variants (CNVs). The prioritization algorithm included the application of several step-by-step custom filters (Figure 2). Finally, variants were classified according to the ACMG (American College of Clinical Genetics) [9].

## 3. Results

We found seven patients with PAH associated with *SOX17* variants. Clinical and molecular features of these patients are described in Table 2 and Appendix A, respectively.

### 3.1. Clinical Description of New PAH Patients with Variants in SOX17

Regarding the onset, five of the patients were pediatric patients. Two were APAH-CHD patients, and three were diagnosed as IPAH patients. These three pediatric idiopathic patients had coincidental defects; patient three (HTP973) and patient six (MSD) had a patent foramen ovale, and patient seven (HTP1031) had a patent foramen ovale and a small patent ductus arteriosus. Regarding the two adult patients, one was associated with connective tissue disease (patient HTP964) and the remaining one was diagnosed with IPAH.

#### 3.1.1. Patient 1

An 11-month-old male was admitted to the intensive care unit, diagnosed with severe bronchiolitis. He had a medical history of choledochal cysts and failure to thrive. He presented with desaturation and respiratory distress, with no response to oxygen and bronchodilator therapy. An echocardiography was performed, showing signs of severe pulmonary hypertension: SPAP 100 mmHg, patent foramen ovale and small patent ductus arteriosus both with right-to-left shunt, and right ventricle dilation and systolic dysfunction. Due to his clinical situation, only non-invasive tests were performed to rule out PH-associated causes. He began treatment of milrinone and epoprostenol continuous infusion. The patient’s condition deteriorated within two days; he suffered cardiac arrest, and he was placed on VA-ECMO. After one month, he underwent lung transplantation. He is currently two years old.

#### 3.1.2. Patient 2

A 2-year-old female with early-onset autism was diagnosed with Eisenmenger syndrome, secondary to a big atrial defect and a patent ductus arteriosus. She was treated with triple sequential combination therapy and experienced multiple episodes of hemoptysis in the course of the disease, dying as a consequence of progressive right heart failure at ten years of age.

#### 3.1.3. Patient 3

An 11-year-old male was diagnosed with IPAH, in functional class III, with frequent syncopal episodes and chest pain episodes. A patent foramen ovale was noted in the transthoracic echocardiography. In the brain MRI, an old ischemic cerebellar lesion was found, without clinical manifestations, and was interpreted as consequence of the right-to-left shunt through the patent foramen ovale. Mild–moderate sleep obstructive apnea was also diagnosed and treatment with nocturnal CPAP was established. He had normal DLCO (DLCO 75%, KCO 85%). He is currently stable, aged 12 years, with quadruple therapy including subcutaneous Treprostinil and Sotatercept^®^ as a participant of the MOONBEAM trial (NCT05587712).

#### 3.1.4. Patient 4

A 34-year-old female was diagnosed with PAH associated with a limiting juvenile rheumatoid arthritis diagnosed in her childhood and treated with corticosteroids and methotrexate. She also had T-cell large granular lymphocyte leukemia and hypothyroidism. Although treated with triple sequential therapy, including subcutaneous Treprostinil, she died due to progressive heart failure whilst being considered for lung transplantation.

#### 3.1.5. Patient 5

A male diagnosed at 3 months of age with severe PAH associated with congenital heart disease (left isomerism, inferior vena cava draining through hemiazygos vein to the coronary sinus and right atrium, and a big atrial septal defect) and high PVR, which precluded the defect closure. Despite triple sequential therapy, the course progressively worsened, with increasing cyanosis and RV dilatation, and he underwent a lung transplant at 17 years of age.

#### 3.1.6. Patient 6

A former smoker, a 54-year-old male was diagnosed with idiopathic PAH. Initially stable under double vasodilator therapy, with ageing, the patient developed atrial fibrillation, obstructive sleep apnea, chronic obstructive pulmonary disease, and severe left ventricle dysfunction; nevertheless, he maintained a precapillary hemodynamic profile. However, it is important to note that these comorbidities have appeared during 20 years of follow-up. Currently, he is 74 years and remains stable with tadalafil therapy as well as with quadruple neurohormonal therapy for left ventricle dysfunction (betablockers, angiotensin receptor neprilysin inhibitor, SGLT2 inhibitor, and mineralocorticoid receptor antagonist) (Figure 3).

#### 3.1.7. Patient 7

A male was diagnosed with IPAH when he was six years old, associated with a patent foramen ovale in functional class II–III, according to the NYHA. He presented positive antinuclear antibodies (ANA title 1/62) without symptoms of rheumatic disease. He is currently stable under triple sequential therapy at ten years age. He showed dilated and tortuous (“cork-screw”) distal pulmonary arteries, evident both in the pulmonary angiography and in the CT scan (Figure 4).

### 3.2. Genetic and Clinical Review of the Total Number of PAH Patients with Variants in SOX17

We detected seven patients with variants in *SOX17*, three with a missense, one with a nonsense variant, and three with a frameshift. In addition, we conducted a review of the previously published variants; some of the clinical features of the patients are summarized in Appendix A. Including our patients, in total, 50 different variants in *SOX17* have been identified in 67 patients. At the molecular level, 22.00% (11/50) are frameshift, 8.00% (4/50) are nonsense, 68.00% (34/50) are missense, and 2.00% (1/50) affect the splicing (Figure 5). These variants were not present in the general pseudonctontrol population databases analyzed (gnomAD exomes, gnomAD genomes, Bravo, 1000 Genomes, ExAC). Most of the missense variants were located within the first exon of *SOX17,* while the LoF variants appeared in the last exon.

At the clinical level, 74.63% (50/67) of the patients had been diagnosed with HPAH/IPAH, 20.89% (14/67) with PAH-CHD, and 4.47% (3/67) with other associated forms of PAH. In our cohort, three out of six patients were diagnosed in childhood, one at the perinatal stage (see Table 2), and the remaining two patients in adulthood. Two of the patients died due to progressive heart failure and two patients underwent lung transplantation. Four out of seven patients received triple combination therapy including systemic prostacyclins.

## 4. Discussion

Nowadays, there are several known potential causative genes associated with PAH (Table 1). In this work, we focused on the relationship between *SOX17* variants and the development of PAH. Here, we presented seven additional PAH patients with variants in this gene and their main clinical features have been summarized. In addition, we have compiled information on previously described variants in this gene as well as certain clinical characteristics of patients, proposing a mutational hot spot associated with PAH and highlighting the importance of the study of these patients in understanding the possible second genetic hits or environmental factors that may be implicated in the phenotypic heterogeneity.

The *SOX17* gene was first associated with PAH patients in a 2018 study whose aim was to identify the missing heritability through whole-genome sequencing in 1038 PAH index cases and 6385 PAH-negative control subjects; it revealed significant overrepresentation in PAH patients of rare variants in some genes such as *SOX17*, among others. Here, we reported seven additional patients with variants in *SOX17* and a review of previously described variants and clinical features from our cohort and from all the individuals reported so far. In total, 67 patients with PAH have been described to have associated variants in *SOX17* since the initial description of the association of the gene with the disease. SOX17 belong to the SRY-box (SOX) family of transcription factors which were firstly described for their role in sex determination and in which its members are now known to be critical regulators of organ development and cell fate decision. The SRY-box family of proteins are characterized by a highly conserved mobility group (HMG)-box DNA-binding domain and include 20 members, almost all showing approximately 50% sequence homology to the HMG-box domain of the SRY (sex-determining region Y) gene [10,11] (Figure 6). *SOX17* has a critical function in embryonic development and plays a crucial role in the development of the cardiovascular system and the remodeling of blood vessels after birth [12,13,14]. It is widely expressed during embryonic development and involved in Wnt/β-catenin and Notch signaling during development [13]. *sox17* mutant mice display heart-looping defects and enlarged cardinal veins during early cardiovascular development [15]. Endothelial-specific deletion of sox17 in embryonic or perinatal mice causes defects in arterial differentiation and vascular formation [16] and regulates endothelial cell proliferation, sprouting, and migration to promote tumor angiogenesis in adult mice [17]. Thereby, dysregulation in *SOX17* expression or activity can contribute to endothelial dysfunction, which eventually may contribute to the development or progression of PAH.

In general, the majority of the patients described in this study have comorbidities or additional conditions. Specifically, 4/7 patients (57%) from our cohort have septal defects, which is in line with the large number of septal defects described by the French group in their cohort of patients [18]. SOX17 expression in the endocardium is crucial for heart development. Deleting SOX17, specifically in the mesoderm, severely impairs endocardium development, affecting cell proliferation and behavior. This leads to impaired cardiomyocyte proliferation, ventricular trabeculation, and myocardium thickening, resulting in abnormal heart morphology likely caused by reduced NOTCH signaling [19]. Concerning the genetic implication of genetics in PAH-CHD patients, although APAH-CHD is an infrequent phenotype for *BMPR2* variant carriers, the significant different *BMPR2* mutation rate between PAH-CHD patients who develop PVD (pulmonary vascular disease) and PAH-CHD patients without PVD demonstrates that the genetic predisposing factor is an important component to evaluate in the process of PVD in PAH-CHD patients [20]. Also, experimental investigations have indicated a correlation between *BMPR2* and the emergence of both simple and complex septal defects [21].

In addition, we identified a patient who had PAH along with a very limiting juvenile idiopathic arthritis (JIA), with a very early onset (5 years), who also presented with large T-cell leukemia. JIA is a heterogeneous group of idiopathic inflammatory arthritis [22]. Some pulmonary complications, such as PAH, are extremely rare features. Spontaneous reports emerged about unusual pulmonary complications in systemic JIA patients that were often fatal, especially PAH [23]. In 2013, Kimura and colleagues published a study featuring 25 patients with systemic JIA and lung disease, which encompassed conditions such as PAH, and the findings of the study indicated that these complications may occur more commonly than previously suspected and are fatal in many patients. Therefore, considering the possible consequences of the appearance of these entities, the genetic study of these patients is very important in order to analyze whether it could be a genetic second hit for an underlying inflammatory autoimmune disease. However, it is very important to clarify that in order to sustain the role of these variants in these phenotypes, it is necessary to study additional patients or to perform functional studies.

Up to 30% of adult and 75% of pediatric PAH cases are associated with congenital heart disease, and the underlying etiology is largely unknown. Research conducted on a group of PAH patients with congenital heart disease indicated that *SOX17* variations are particularly prevalent in this subgroup, which experiences an early onset of the disease. Here, we presented a pediatric patient with a novel variant who showed the same tortuous arteries described by the French group in 81% (13/16) of their patients [18], a finding that could be associated with the also mentioned high incidence of hemoptysis. In our cohort, only one patient experienced multiple episodes of severe hemoptysis, although collateral vessels could not be demonstrated in that case. In addition, we reported a pediatric patient carrying a pathogenic variant not described previously and who presented with the same corkscrew tortuous arteries, suggesting this observed relationship in that case. Finally, we described patient seven, who had a choledochal cyst (Table 2). It should be noted that SOX17 regulates the differentiation and maintenance of the biliary phenotype and functions, and therefore, this fact highlights its involvement.

Pathogenic and likely pathogenic variants in *SOX17* have been identified in patients with familial or idiopathic PAH [24,25,26,27]. These variants are inherited in an autosomal dominant manner. Additional studies have confirmed the genetic association between *SOX17* and H/IPAH in many populations with diverse genetic backgrounds [6,8,27]. In addition to *SOX17*, almost 30 genes (Table 1) have been associated with the development of the disease and can be also associated with the highly heterogeneous clinical manifestations. However, collectively these analyses have identified 60 patients with variants in *SOX17,* including missense, nonsense, frameshift, and splicing variants, in patients with H/IPAH and other associated forms of PAH. We have added seven new cases (four IPAH, one APAH-CTD, and two APAH-CHD), of which two of the IPAH patients showed the same variant, *SOX17*:NM_022454.4:c.499_520del:p.Leu167TrpfsTer213, previously reported in other patients in the literature. In total, nine PAH patients have been reported as having the same variant, suggesting that this variant is a clear hot spot associated with IPAH. This very frequent change does not affect the region that encodes a functional domain of the protein, nor is it a highly conserved region according to in silico predictors (PhyloP score); however, it could be altering the final protein structure. Functional studies will be needed to determine the effect of this variant. In addition, it is important to highlight that two of the nine patients with the variant p.Leu167TrpfsTer213 were diagnosed with PAH-CHD and the rest as IPAH. This could suggest that it is possible for other genetic events or even environmental factors to be implicated in these differences. Thus, further studies are needed to clarify these phenotypic differences, even though the patients have the same variant.

*SOX17* is organized into two exons. The protein encoded by *SOX17* possesses a DNA-binding HMG box at the N-terminal and β-catenin-binding domains at the C-terminal (Figure 7). We also highlight that several nonsense and frameshift pathogenic variants found in PAH patients occur in the last exon (Figure 7), specifically in the conserved β-catenin-binding domain. Consequently, these variant transcripts are expected to evade the process of nonsense-mediated decay, resulting in the production of a malfunctioning protein and the haploinsufficiency of the protein. Meanwhile, most missense variants are in exon 1, in the HMG-box domain, and some of them, 50% (7/14), affect highly conserved residues in all isoforms (Figure 7), which could support their possible deleterious role. In this sense, it would be necessary to perform more studies to explain the functional consequences of these variants at the protein level.

Regarding the variant Leu167TrpfsTer213, it is not within a conserved functional or other preserved domain (Figure 6 and Figure 7). Nonetheless, the protein will be truncated prior to the B-catenin binding domain, maybe deactivating SOX17-mediated regulation of B-catenin expression. This deactivation can lead to abnormal vasculature remodeling, indicating a clear pathogenic variant causing altered protein function. Therefore, experimental data are necessary to determine the impact of the resulting protein. Identifying more patients with variants in *SOX17* enables prognosis and associated condition studies, informing clinical care. Recurrent variants further enhance our ability to classify these variants clinically and describe the natural disease progression.

Wang et al. [28] reported the presence of a nonsense variant in the terminal exon of *SOX17* within a family spanning multiple generations. In vitro luciferase assays demonstrated that the variant allele led to a 14-fold reduction in the activity of the target gene NOTCH1 reporter and the repression of β-catenin, compared to the wild-type allele. The proper functioning of NOTCH1 is crucial for the regeneration of endothelial cells in response to injury, and the deletion of Notch1, specifically in endothelial cells in mice, has been shown to worsen PAH [29]. When the expression of β-catenin is unrestrained, it promotes abnormal remodeling of blood vessels. Immunolocalization studies have clearly established that *SOX17* is specifically expressed in the endothelial cells of the pulmonary arterioles in both healthy individuals and patients with vascular lesions [6].

In summary, here, we have described seven new patients with variants in *SOX17*, a definitively associated gene with PAH, according to ClinGen classification [7]. We have proposed a mutational hot spot because 9/67 patients had the same variant. However, it would be necessary to extend our studies to explain why there are phenotypic differences between these nine patients (two were diagnosed with PAH-CHD and seven with IPAH/HPAH), despite having this same genotype. In addition, we have added evidence to the hypothesis that variants in this gene are associated with APAH-CHD as we have described two new patients with this diagnosis and specifically, with tortuous arteries. We conclude that genetics may be a very important factor implicated in the development of PAH, but associated conditions can facilitate the onset of pulmonary vascular disease: congenital heart disease, autoimmune background in connective tissue diseases.

Further research is needed to fully understand the specific mechanisms through which *SOX17* and PAH are connected. However, the identification of this association suggests that SOX17 could be a potential target for therapeutic interventions in PAH. Modulating SOX17 expression or activity could potentially help to restore normal endothelial function and improve pulmonary arterial health in individuals with PAH. It is important to note that while there is evidence of a connection between *SOX17* and PAH, the research in this area is still evolving, and there may be additional factors and molecular pathways involved in the development and progression of PAH. While progress has been made in understanding *SOX17* variants and their relationship to PAH, further research is still needed to clarify the precise mechanisms and extent of this association. The genetics of PAH are complex, and *SOX17* mutations appear to play a role in only a specific subset of patients.

## Figures and Tables

**Figure 1 genes-14-01965-f001:**
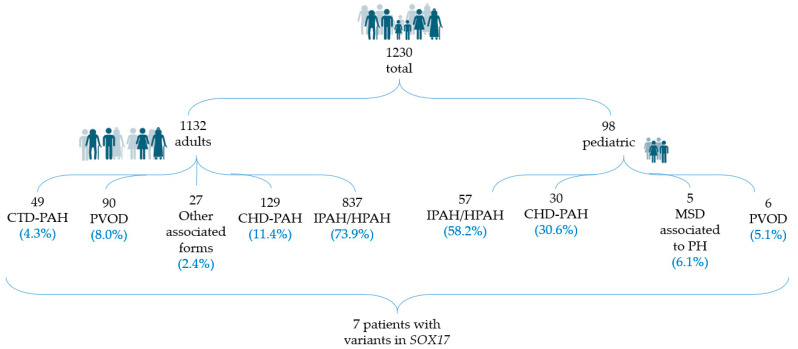
Description of the total cohort analyzed in this study.

**Figure 2 genes-14-01965-f002:**
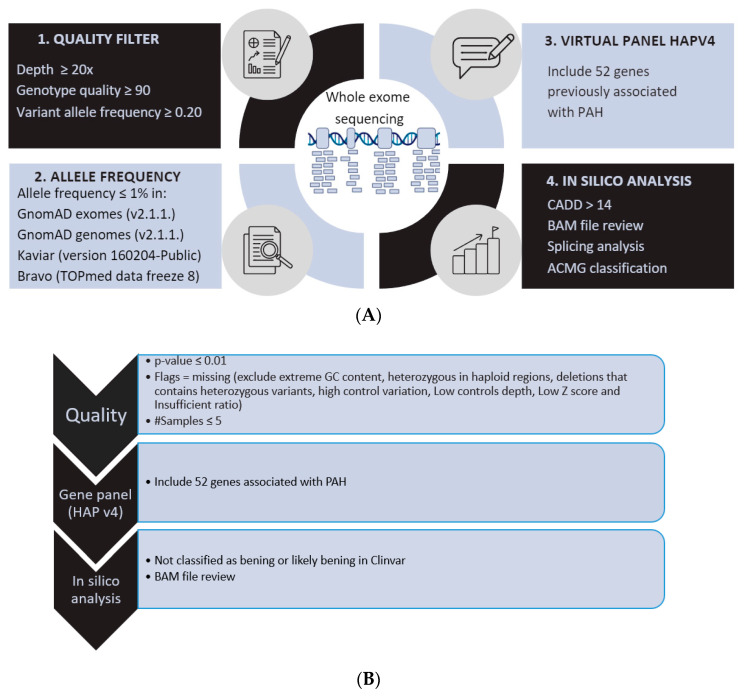
(**A**) Workflow for single nucleotide variant and insertion-deletions variant prioritization. Schematic representation of the pipeline for tertiary/prioritization analysis. BAM review was performed through Alamut and IGV; splicing analysis was performed through SpliceSiteFinder-like, MaxEntScan, GeneSplicer, and NNSPLICE. (**B**) Workflow for copy number variant prioritization. #Samples: Patients were divided into different work pools (minimum 20 patients) and analyzed in the same project. In this filter it is specified that the variant has to be found in 5 or less patients of that pool.

**Figure 3 genes-14-01965-f003:**
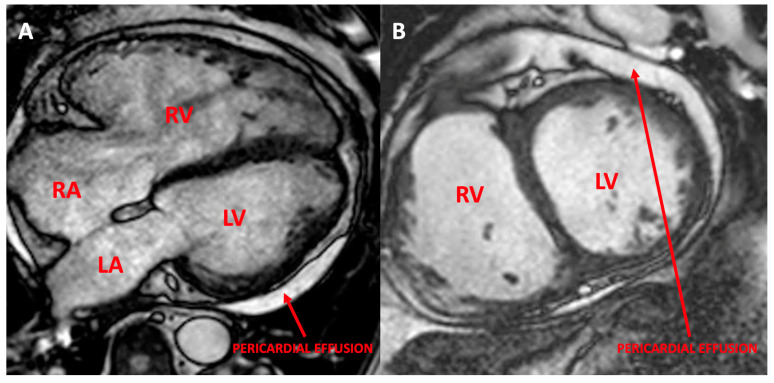
Cardiac magnetic resonance imaging of the first patient. (**A**) A 4-chamber CINE image showing the predominant right chamber dilation as compared to the left ventricle; (**B**) a short-axis CINE sequence demonstrating the severe dilation of both ventricles, with a centered interventricular septum. Pericardial effusion could also be noted. RV: right ventricle; RA: right atrium; LV: left ventricle; RV: right ventricle.

**Figure 4 genes-14-01965-f004:**
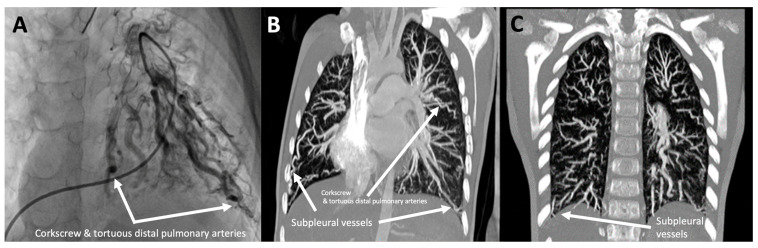
Angiographic study of patient HTP973. (**A**) Subselective pulmonary angiography of the lower right anterior pulmonary trunk, showing generalized “corkscrew”, tortuous subsegmental vessels. (**B**,**C**) Computed tomography angiography images demonstrating the presence of “corkscrew” and tortuous arteries in both lungs, as well as subpleural vessels.

**Figure 5 genes-14-01965-f005:**
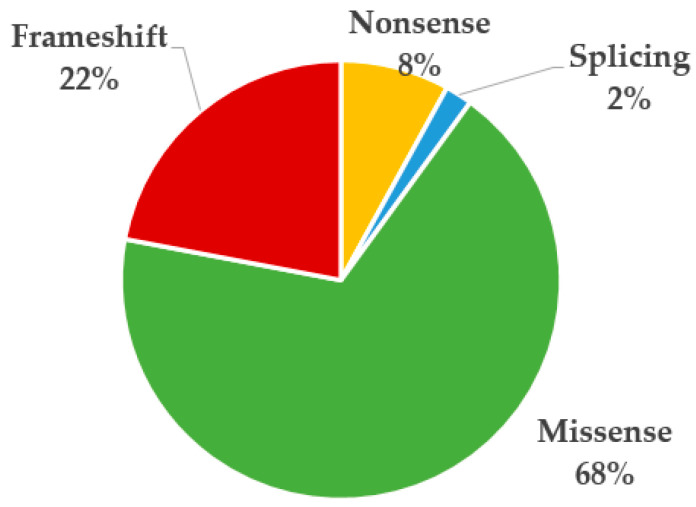
Proportion of variants detected in *SOX17* according to the variant type in previously published patients and our cohort.

**Figure 6 genes-14-01965-f006:**
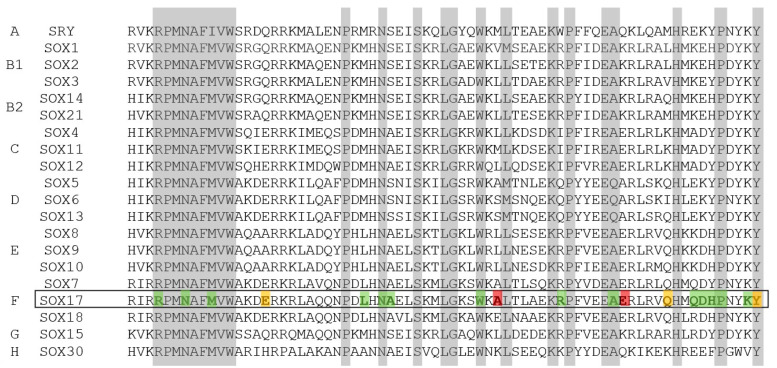
Amino acid sequence of HMG-box domain in SOX family. The SOX family is comprised of 20 members and is grouped based on domain organization. Each SOX family member contains a highly conserved HMG domain (in grey, the most conserver residues). Variants detected in *SOX17* in PAH patients are shown in colors: green (missense), yellow (nonsense), and red (frameshift).

**Figure 7 genes-14-01965-f007:**
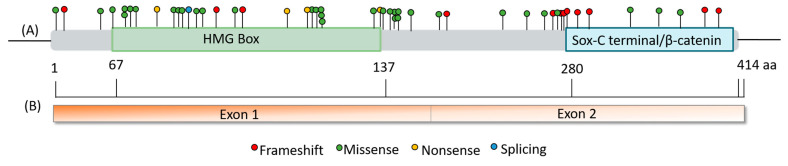
Summary and representation of the variants found in *SOX17*. (**A**) Schematic representation of the *SOX17*-encoded protein and location of genetic variants according to the effect of the variant in the protein; (**B**) linear schematic of the *SOX17* gene.

**Table 1 genes-14-01965-t001:** Causal and candidate genes associated with PAH.

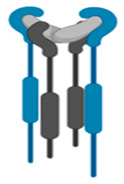 BMP/TGFβ	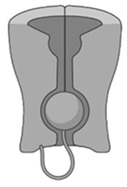 Channels	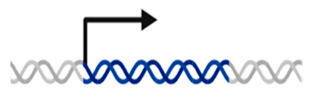 Transcription Factors	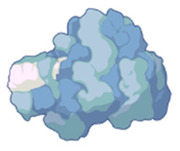 Other
*ACVRL1* *BMPR2* *ENG* *GDF2* *SMAD9* *CAV1* *BMP10* *BMPR1A* *BMPR1B* *SMAD1* *SMAD4*	*ATP13A3* *KCNK3* *ABCC8* *AQP1*	*EIF2AK4* *SOX17* *TBX4* *KLF2*	*KDR* *TET2* *GGCX* *FBLN2* *KLK1* *PDGFD* *NOTCH3*

**Table 2 genes-14-01965-t002:** Clinical and genetic information of PAH patients with SOX17 variants. 6MWT (six-minute walk test); ANA (antinuclear antibodies); ASD (atrial septal defect); AT (atrial tachycardia); y.o (years old); LVSD (left ventricle systolic dysfunction); MPAP (mean pulmonary artery pressure); NYHA (New York Heart Association); OSA (obstructive sleep apnea); PAWP (pulmonary artery wedge pressure); PDA (patent ductus arteriosus); PFO (patent foramen ovale); PVR (pulmonary vascular resistance); RA (rheumatoid arthritis); RHC (right heart catheterization); VA-ECMO (venoarterial extracorporeal membrane oxygenation). * Tadalafil for PAH as pulmonary vasodilator. Quadruple combination therapy for severe left ventricle dysfunction (Empagliflozin, Bisoprolol, Sacubitril/Valsartan, and Spironolactone); ** iPDE5, ERA, and SC Treprostinil; *** unstable at diagnosis. Transthoracic echocardiography showed signs of severe pulmonary hypertension (SPAP 63 mmHg, suprasystemic pressure) and severe right ventricle dilation and dysfunction (TAPSE 13 mm). The clinical information of the seven patients identified in our cohort is described below. § Risk assessment in pediatric patients is not validated.

Patient ID	HTP1301 Patient 1	HTP973 Patient 2	HTP1089 Patient 3	HTP964 Patient 4	HTP1010 Patient 5	HTP379 Patient 6	SPO Patient 7
Sex	Male	Male	Male	Female	Female	Male	Male
PAH diagnosis	IPAH	IPAH	APAH-CHD	APAH-CTD	APAH-CHD	IPAH	IPAH
Age at diagnosis (years)	0.9 (11 months)	6	0.2 (3 months)	34	2	54	11
NYHA	IV	III	II-III	IV	III	II	III
Family history	No	No	No	No	No	No	No
**Genetic information**							
Genomic position	8:55370907	8:55370916	8:55370942	8:55371799	8:55371798	8:55371798	8:54459532
* cDNA position (NM_022454.4)	c.209G>T	c.218A>C	c.244G>T	c.489G>C	c.499_520del	c.499_520del	c.788dup
Protein position	p.Arg70Leu	p.Asn73Thr	p.Glu82Ter	p.Gln163His	p.Leu167TrpfsTer213	p.Leu167TrpfsTer213	p.Glu264Glyfs101
**Hemodynamic parameters at diagnosis**							
MPAP (mmHg)	No RHC	52	52	53	No RHC ***	56	68
PAWP (mmHg)	No RHC	5	12	14	No RHC	12	9
PVR (Wood units)	No RHC	16.0	16.5	15.6	No RHC	6.96	17.6
6MWT (meters)	No 6MWT	530	545	393	No 6MWT (developmental delay)	580	450
Associated conditions	Choledochal cyst PFO Small PDA	PFO Positive ANA	Left isomerism, inferior vena cava draining through hemiazygos vein to coronary sinus and right atrium ASD	Juvenile RA T-cell large granular lymphocyte leukemia Hypothyroidism	Big ASD and PDA in Eisenmenger Autism	Severe LVSD and AT at the age of 67	PFO Hypercoagulability OSA
**Treatment and follow-up**							
§ Baseline risk (according to the 2022 ESC/ERS Guidelines)	-	-	-	Intermediate risk	-	Intermediate risk	-
§ Outcomes and current risk according to the 4-strata 2022 ESC/ERS risk score at follow-up	VA-ECMO Lung transplantation	Death	Alive	High risk. Dead at 35 years despite triple sequential therapy including systemic prostacyclins	Lung transplant at age 17 years	Intermediate–low risk under monotherapy with PDE5i	Alive
Last known PAH therapies	Epoprostenol	** Triple sequential therapy	** Triple sequential therapy	** Triple sequential therapy	** Triple sequential therapy	* Tadalafil	Quadruple therapy: Tadalafil, macicentan, Sc treprostinil + sotatercept

## Data Availability

The data are not publicly available due to privacy.

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
