# Peer review of "Seven Additional Patients with SOX17 Related Pulmonary Arterial Hypertension and Review of the Literature"

_genes, 2023, doi:10.3390/genes14101965_

Round 1

Reviewer 1 Report

Gallego-Zazo and colleagues report the identification of seven Spanish pulmonary arterial hypertension patients with rare variants in definitive PAH causal gene, SOX17. Enrichment of SOX17 variants has been previously reported for both pediatric- vs adult-onset PAH and PAH associated with congenital heart disease vs PAH alone. Five of the new Spanish cases are pediatric and two of these have PAH-CHD. Finally, two of the new patients have a recurrent SOX17 variant previously reported in seven PAH patients, further establishing the pathogenicity of this frameshift variant. The manuscript has several weaknesses including inadequate description of methods, data/details missing from tables/figures, organization of manuscript and Table 1, and overall length of manuscript.

Major comments

1. General comment – The manuscript text should be edited for brevity. There is a lot of redundancy in the text, including paragraphs in the introduction that are reiterated in the discussion.

2. Methods, patients – State the # of patients included from each cohort, the inclusion/exclusion criteria, and specifically whether patients with pathogenic/likely pathogenic variants in known PAH causal genes were included/excluded. 

3. Methods, genetic analysis – There is inadequate description of the exome sequencing data processing/quality control filtering. If this has been described previously for both cohorts, then provide the references. Variant prioritization is inadequately described as well. At a minimum, state the human reference genome build used, the allele frequency threshold applied, and how the missense variants were filtered for deleteriousness (i.e. CADD >20, REVEL = ?, etc). Was the variant screen limited to autosomal dominant alleles? The text indicates that variant prioritization included screening for copy number variants. However, CNVs cannot be detected from exome sequencing data. Was another method used or is this errant text?

4. Results – This section needs reorganization to distinguish new data from the Spanish cohorts from previously reported data. For example, table 1, figure 2, and figure 3 should be presented before figure 1 and renumbered accordingly.

5. Results, table 1 – Add a column for the genetic variants. Provide genome coordinates, transcript ID, coding and protein nomenclature (as in suppl table 1). Suggest organizing the table according to variant position (i.e. list the patient with the most 5’ variant first and ending with the most 3’ variant). In this way, recurrent variants and variants in the same conserved domains or exons can be identified readily. RHC diagnostic data is provided. Is there any follow-up data that could inform prognosis for PAH patients with SOX17 variants? For the associated phenotypes row, suggest expanding the lists for consistency between text and table, and using abbreviations (i.e LV dysfunction, juvRA, TLGL, etc) to accommodate the additional phenotypes. For the TLGL listing, the year (of diagnosis?) is provided but it would be better to indicate the patient age at diagnosis.

6. Results, figures 2 and 3 – Add labels or arrows to indicate the RV, LV, pericardial effusion, and the affected vessels in 3.

7. Results, “global cohort” – Define the term “global cohort.” Add references for the cohorts included. Describe the cohort stating the total number of PAH patients and the percent I/HPAH, APAH-CHD, and other PAH. Have the variant coordinates all been lifted over to the same genome build? Have you confirmed that the SOX17PAH variant carriers are unrelated (i.e no one reported/counted twice)? For figure 1 (which should be moved to figure 3), consider changing “protein effect” to “variant type.” Recommend removing panel B as it could be misleading. Figure 1B makes it look like SOX17 variants explain more of I/HPAH than PAH-CHD but the opposite is true. It would be more informative to show the percentage of global cohort APAH-CHD, I/HPAH, and other PAH explained by the variants; in this way, the biggest portion of pie would be for APAH-CHD. For the percentages listed in the figures, reduce to one or no significant figure(s).

8. Discussion – a) There is redundant information in the introduction, results, and discussion. The second, and perhaps the third, paragraph (discussion of known and candidate PAH causal genes) belongs in the introduction only. b) There are sentences highlighting BMPR2 as an APAH-CHD gene. However, 4/180 translates to 2% of BMPR2 variant carriers have diagnoses of APAH-CHD, indicating that APAH-CHD is a rare phenotype for BMPR2 variant carriers. c) Phenotypes clearly associated with SOX17 variants include PAH and CHD; additional phenotypes observed in single variant carriers can be mentioned but require study of additional patients or functional data to infer causation by SOX17. d) It is stated that Leu167TrpfsTer213 is not located in a conserved functional domain or other conserved domain, so experimental data would be needed to determine the effect of the resulting protein. However, the protein will truncate upstream of the B-catenin binding domain, inactivating SOX17-mediated regulation of B-catenin expression which leads to abnormal remodeling of vasculature. It is clearly a pathogenic variant leading to altered protein function.

The take-home message should be concise and clear: The clinical implications of identifying more SOX17 PAH variant carriers is to allow studies of prognosis and associated conditions to inform clinical care. For recurrent variants, there is an even greater ability to clinically classify the variants and to describe the natural course of disease.

9. Table 2 – This table does not seem relevant to this report. However, if included, the title should be changed as the genes shown include both known causal and candidate genes, not just candidate genes. There is a typo in SMAD9. Two of the included genes have been reputed recently.

10. Figure 4 – If this figure is limited to the HMG box of SOX family members, this should be stated explicitly, most likely in the title. Perhaps variants at the most highly conserved residues (shaded with gray) deserve specific mention in the discussion and stating whether any of these are recurrent variants? In the text, specifically mention the variants with coding and/or protein nomenclature (3 aa abbreviations as in supple table 1).

11. Figure 5 – Where are the recurrent variants? Some are shown but there should be more in the HMG box. Also, the putative 9 recurrences of Leu167Trp variant are missing. In B, what is the significance of aa 67? Residue 137 is shown but not 167. Is this a typo? How many nucleotides upstream of exon1-exon2 junction is c.499-520del (p.Leu167Trp)? Is it predicted to evade NMD?

Minor comments

1. Suppl table 1 – The “age of diagnosis” column is not aligned with the rest of the row.

2. Has reference 19 been peer-reviewed and published since the 2019 biorxiv publication?

The manuscript could use minor editing for English language.

Reviewer 2 Report

The possible connection of sox17 loss of function due to mutation (frameshift, non-sense or missense) has been emerging as an important area to target sox17 for therapeutic or genetic modification to attenuate the pathologic complications of PAH. Since PAH is considered as a rare disease with no permanent cure, finding additional patients that falls under PAH classification is certainly important to understand the etiology of the disease and possible mechanisms associated with either onset or progression of the disease in relation to certain  molecular descriptors. Here the authors have The study took advantage of selecting both pediatric (98) and adult cohort (1132) of patients with sox17 pathogenic variant from the Spanish PAH registry (REHAP) and hypertension registry (REHIPED) and upon systemic classification of the clinical parameters following established diagnostic guidelines were able to identify 3 out of 98 (3.06%) and 4 out of 1132 (0.03%) pediatric and adult PAH patients respectively. Along with newly identified 7 patients the current study provides a summary and review of the molecular parameters for the PAH subjects previously identified and described in the published literature. The study reports appropriate ethical approvals and informed consent statement obtained from the participating patients.

1.       Page 3, line 106: The correct number would be ‘three out of 98….’

2.       The study does not report the time span of data collection (start and end of the study). Is the study still ongoing with the patients who are still alive?

3.       Was there any identifiable phenotypic presentation of the sox17 pathogenic variant in patients with PAH and non-PAH groups?

4.       Table 1 reports the family history of the 7 PAH patients with sox17 pathogenic variant but this information is not adequately describing what family history has been referenced here? Was that the identification of sox17 mutation in either or both parents? Was there any information available on the sox17 genetic makeup of the parents or any phenotypic documentations available for them?

5.       Matrix remodeling involving matrix metalloproteinases (mainly MMP2 and 9) has long been identified as a pathologic hallmark in pulmonary artery of the PAH patients. Was there any data or indication available in the current pediatric and adult PAH cohorts on the presence of specific sox17 mutations and the MMP gene or protein expression status?

6.       What was the origin of sox17 mutation that is thought to be linked to PAH. Was it genetically inherited or spontaneous? From Table 1, it is evident that that the age of PAH diagnosis and the life expectancy largely differ. Was there any plausible explanation or data available to address that issue? 6 out of 7 PAH patients (~86%) with sox17 mutations are male. Does that gender dependent prevalence of the PAH cases agree well with the past data found in literature?

Author Response

Please see the atachment
